# Measuring and Controlling Solution Degeneracy across Task-Trained Recurrent Neural Networks

**Ann Huang**[1,2,3]**, Satpreet H. Singh**[2,3]**, Flavio Martinelli**[2,3,4]**, Kanaka Rajan**[2,3]

[1]Harvard University    [2]Harvard Medical School    [3]Kempner Institute    [4]EPFL

`annhuang@g.harvard.edu`

## Abstract

Task-trained recurrent neural networks (RNNs) are widely used in neuroscience and machine learning to model dynamical computations. To gain mechanistic insight into how neural systems solve tasks, prior work often reverse-engineers individual trained networks. However, different RNNs trained on the same task and achieving similar performance can exhibit strikingly different internal solutions, a phenomenon known as solution degeneracy. Here, we develop a unified framework to systematically quantify and control solution degeneracy across three levels: behavior, neural dynamics, and weight space. We apply this framework to 3,400 RNNs trained on four neuroscience-relevant tasks—flip-flop memory, sine wave generation, delayed discrimination, and path integration—while systematically varying task complexity, learning regime, network size, and regularization. We find that higher task complexity and stronger feature learning reduce degeneracy in neural dynamics but increase it in weight space, with mixed effects on behavior. In contrast, larger networks and structural regularization reduce degeneracy at all three levels. These findings empirically validate the Contravariance Principle and provide practical guidance for researchers seeking to tune the variability of RNN solutions, either to uncover shared neural mechanisms or to model the individual variability observed in biological systems. This work provides a principled framework for quantifying and controlling solution degeneracy in task-trained RNNs, offering new tools for building more interpretable and biologically grounded models of neural computation.

## 1 Introduction

Recurrent neural networks (RNNs) are widely used in machine learning and computational neuroscience to model dynamical processes [1, 2, 3, 4, 5, 6]. Traditionally, the study of task-trained RNNs has focused on reverse-engineering a single trained model, implicitly assuming that networks trained on the same task would converge to similar solutions—even when initialized or trained differently. However, recent work has shown that this assumption does not hold universally, and the solution space of task-trained RNNs can be highly degenerate: networks may achieve the same level of training loss, yet differ in out-of-distribution (OOD) behavior, internal representations, neural dynamics, and connectivity [7, 8, 9, 10, 11].

These raise fundamental questions about the solution space of task-trained RNNs: **What factors govern the solution degeneracy across independently trained RNNs?** Despite extensive work in feedforward networks showing how different initializations and stochastic training can yield divergent solutions, RNNs still lack a systematic and unified understanding of the factors that govern solution degeneracy [12, 13, 14, 15, 16, 17, 18, 19, 20]. Cao and Yamins [21] proposed the *Contravariance Principle*, which posits that as the computational objective (i.e., the task) becomes more complex, the solution space should become less dispersed. While this principle is intuitive and compelling, it has thus far remained largely theoretical and has not been directly validated through empirical studies.

In this paper, we introduce a unified framework for quantifying solution degeneracy at three levels: behavior, neural dynamics, and weight space. As illustrated in Figure 1 , we quantify degeneracy across

Preprint. Under review.

behavior, dynamics, and weights, and examine how it is shaped by four key factors. Leveraging this framework, we isolate four key factors that control solution degeneracy—task complexity, learning regime, network width, and structural regularization. By systematically varying task complexity, learning regime, network width, and regularization, we map how each factor shapes degeneracy across behavior, dynamics, and weights.

We find that as task complexity increases—whether via more input–output channels, higher memory demand, or auxiliary objectives—or as networks undergo stronger feature learning, their neural dynamics become more consistent, while their weight configurations grow more variable. In contrast, increasing network size or imposing structural regularization during training reduces variability at both the dynamics and weight levels. At the behavioral level, each of these factors reliably modulates behavioral degeneracy; however, the relationship between behavioral and dynamical degeneracy is not always consistent.

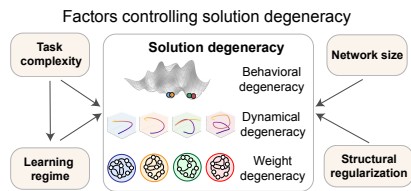

Figure 1: Key factors shape degeneracy across behavior, dynamics, and weights.

## 2 Methods

**Model architecture and training.** We use discrete-time nonlinear *vanilla* RNNs with update $\mathbf{h}_t = \tanh\left(\mathbf{W}_h \mathbf{h}_{t-1} + \mathbf{W}_x \mathbf{x}_t + \mathbf{b}\right)$ where $\mathbf{h}_t \in \mathbb{R}^n$ is the hidden state, $\mathbf{x}_t \in \mathbb{R}^m$ is the input, $\mathbf{W}_h \in \mathbb{R}^{n \times n}$ and $\mathbf{W}_x \in \mathbb{R}^{n \times m}$ are the recurrent and input weight matrices. A linear readout maps $\mathbf{h}_t$ to outputs. Networks are trained with BPTT (Adam optimizer, no weight decay) [22]. For each task, we train 50 seeds with 128 hidden units, initializing $\mathbf{W}_h, \mathbf{W}_x \sim \mathcal{U}(-1/\sqrt{n}, 1/\sqrt{n})$. Training continues until networks reach a near-asymptotic training loss threshold, after which we allow 3 epochs' patience period and stop training to assess degeneracy across solutions (Appendix G).

**Tasks.** We evaluate four neuroscience-relevant tasks eliciting distinct dynamics: pattern recognition (N-Bit Flip-Flop), delayed decision-making (Delayed Discrimination), pattern generation (Sine Wave Generation), and evidence accumulation (Path Integration). Task details and example neural dynamics required to solve the tasks are in Appendix A and F .

**Degeneracy metrics.** *Behavioral degeneracy* measures the variability in network responses to out-of-distribution (OOD) inputs. We measure OOD performance as the mean squared error of all converged networks under a temporal generalization condition (double the delay for Delayed Discrimination; double the trial length otherwise). Behavioral degeneracy is the standard deviation of the OOD losses.

*Dynamical degeneracy* quantifies the average pairwise difference in networks' neural dynamics through Dynamical Similarity Analysis (DSA) [23]. DSA compares the topological structure of dynamical systems and has been shown to be more robust to noise and better at identifying behaviorally relevant differences than prior metrics such as Procrustes Analysis and Central Kernel Alignment [24]. For a pair of networks $X$ and $Y$, DSA identifies a linear forward operator for each system—$A_x$ and $A_y$—which maps neural activity from one time step to the next. These operators are then compared up to a rotation. The DSA distance between two systems is computed by minimizing the Frobenius norm between the operators, up to rotation: $d_{\mathrm{DSA}}(A_x, A_y) = \min_{C \in O(n)} \left\| A_x - C A_y C^{-1} \right\|_F$, where $O(n)$ is the orthogonal group. We define dynamical degeneracy as the average DSA distance across all network pairs. Additional details are provided in Appendix I.

*Weight degeneracy* is defined as a permutation-invariant Frobenius distance between recurrent weights $d_{\mathrm{PIF}}(\mathbf{W}_1, \mathbf{W}_2) = \min_{\mathbf{P} \in \mathcal{P}(n)} \left\| \mathbf{W}_1 - \mathbf{P}^\top \mathbf{W}_2 \mathbf{P} \right\|_F$, normalized by parameter count when comparing different widths (Appendix I.2).

## 3 Results

### 3.1 Task complexity modulates degeneracy across levels

We varied task complexity by increasing the number of independent input–output channels of each task, which effectively duplicated the task across dimensions and increased the representational load of networks by forcing them to multitask. Higher task complexity constrains the space of viable dynamical solutions, leading to tighter clustering and greater similarity across independently trained networks (Fig. 2AB). At the behavioral level, networks trained on more complex tasks

consistently showed greater consistency and lower variability in their responses to OOD test inputs (Fig 2D). Together, the results at the behavioral and dynamical levels support the *Contravariance Principle*, which posits an inverse relationship between task complexity and the dispersion of network solutions [21]. At the weight level, however, we found that pairwise distances between converged RNNs' weight matrices increased consistently with task complexity (Figure 2C), which likely reflects increased dispersion of local minima in weight space for harder tasks [25, 26, 27, 28, 29, 30, 31]. In Appendix B, we explore two alternative approaches of varying task complexity: increasing the task's memory demand and adding auxiliary objectives. We find that the trends in solution degeneracy hold consistently across these approaches.

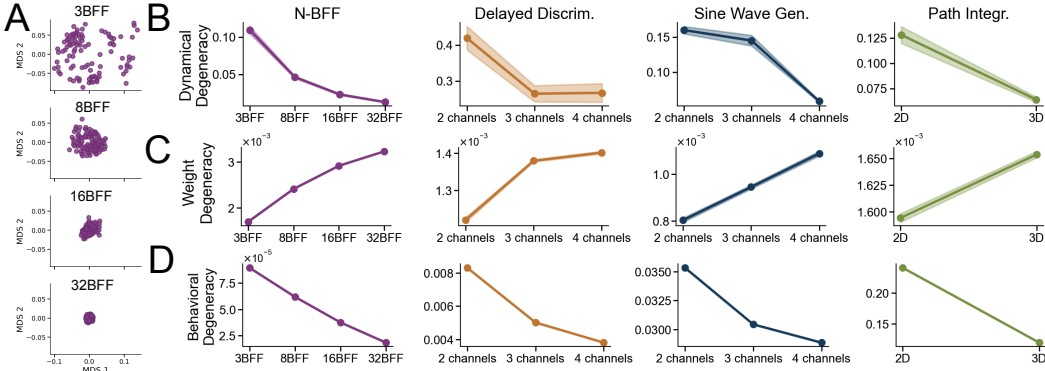

Figure 2: **Higher task complexity reduces dynamical and behavioral degeneracy, but increases weight degeneracy. (A)** Two-dimensional MDS embedding of network dynamics shows that independently trained networks converge to more similar trajectories as task complexity increases. **(B)** Dynamical, **(C)** weight, and **(D)** behavioral degeneracy across 50 networks as a function of task complexity. Shaded area indicates $\pm 1$ standard error.

## 3.2 Controlling feature learning reshapes degeneracy across levels

In deep learning theory, neural networks can operate in either a lazy or rich learning regime [32, 33, 34, 35]. In the lazy regime, weights and internal features remain largely unchanged during training. In the rich (feature learning) regime, networks reshape their hidden representations and weights to capture task-specific structure [32, 36, 37, 33].

Intuitively, when networks undergoes strong feature learning, they converge to more consistent task-specific neural dynamics, leading to lower dynamical degeneracy. To causally test whether feature learning affects solution degeneracy, we used a principled parameterization known as maximum update parameterization ($\mu P$), where a single hyperparameter—$\gamma$—controls the strength of feature learning: higher $\gamma$ values induce a richer feature-learning regime [35, 32, 34, 33]. More specifically, the network output is scaled as

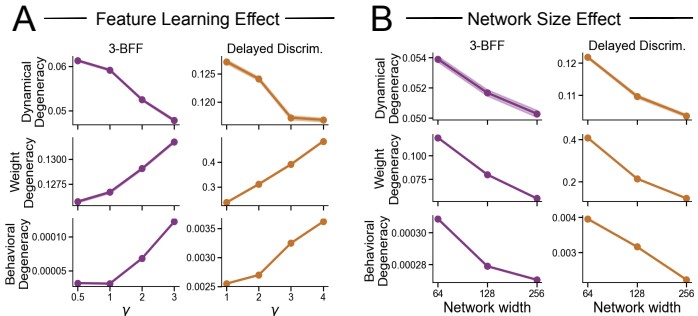

Figure 3: **(A) Stronger feature learning reduces dynamical degeneracy but increases weight and behavioral degeneracy. (B) Larger networks reduce degeneracy across weight, dynamics, and behavior.** Panels show degeneracy at the dynamical, weight, and behavioral levels (top to bottom). Shaded area indicates $\pm 1$ standard error.

$f(t) = \frac{1}{\gamma N} W_{\text{readout}} \phi(h(t))$. A detailed explanation of $\mu P$ and its relationship to the standard parameterization is in Appendix L and M. For each task, we trained networks with multiple $\gamma$ values and confirmed that larger $\gamma$ consistently induces stronger feature learning(Appendix N).

We observed that stronger feature learning reduced degeneracy at the dynamical level but increased it at the weight level (consistent for all four tasks, see Appendix D). This finding aligns with prior work in feedforward networks, where feature learning was shown to reduce the variance of the neural tangent kernel across converged models [38]. Notably, stronger feature learning was shown to push networks to travel farther from their initialization [39, 36], resulting in more dispersed final weights and higher weight degeneracy. At the behavioral level, however, increasing feature-learning strength leads networks to overfit the training distribution (Appendix K.2). We hypothesize that stronger feature learning exacerbates overfitting, increasing both average OOD loss and the variability of OOD behavior across models (Figure 8) [40, 41, 42, 43].

### 3.3 Larger networks yield more consistent solutions across levels

Although larger networks may yield more consistent solutions via self-averaging and improved convergence [44, 45, 46, 47, 48], this outcome is not guaranteed without controlling for feature learning, as increasing network width pushes models towards the lazy regime, where feature learning is suppressed [49, 37, 32, 33, 34]. To disentangle these competing effects, we again use the $\mu P$ parameterization, which holds feature learning strength constant (via fixed $\gamma$) while scaling width. Across all tasks, larger networks consistently exhibit lower degeneracy at the weight, dynamical, and behavioral levels, producing more consistent solutions across random seeds (Figure 9; results hold consistent for all four tasks, see Appendix E). This pattern aligns with findings in vision and language models, where wider networks converge to more similar internal representations [50, 51, 52, 53, 54].

### 3.4 Structural regularization reduces solution degeneracy

Low-rank and sparsity constraints are widely used structural regularizers in neuroscience-inspired modeling and efficient machine learning [4, 55, 56, 57, 58]. A low-rank penalty compresses the weight matrices into a few dominant modes, while an $\ell_1$ penalty drives many parameters to zero and induces sparsity. In both cases, task-irrelevant features are pruned, nudging independently initialized networks toward more consistent solutions on the same task. To test this idea, we augmented the task loss with either a nuclear-norm penalty on the recurrent weights $\mathcal{L} = \mathcal{L}_{\text{task}} + \lambda_{\text{rank}} \sum_{i=1}^{r} \sigma_i$, where $\sigma_i$ are the singular values of the recurrent matrix, or an $\ell_1$ sparsity penalty: $\mathcal{L} = \mathcal{L}_{\text{task}} + \lambda_{\ell_1} \sum_i |w_i|$. We focused on the Delayed Discrimination task to control for baseline difficulty, but both regularizers consistently reduced degeneracy across all levels—and similar effects hold in other tasks (Appendix P, Figure 4).

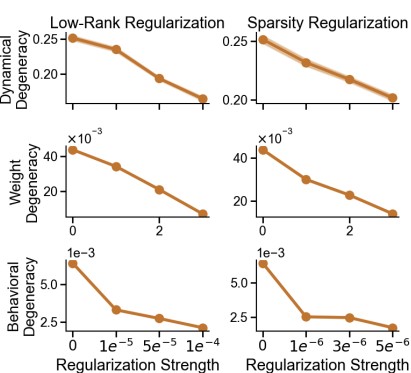

Figure 4: **Low-rank and sparsity regularization reduce solution degeneracy across all levels.** Shaded area indicates $\pm 1$ standard error.

## 4 Discussion

Table 1: Summary of how each factor affects solution degeneracy. Arrows indicate the direction of change for each level as the factor increases. Contravariant factors shift **dynamic** and **weight degeneracy** in opposite direction; covariant factors shift them in the same directions.

| Factor | Dynamics | Weights | Behavior |
|---|---|---|---|
| **Higher Task complexity (contravariant)** | ↓ | ↑ | ↓ |
| **More Feature learning (contravariant)** | ↓ | ↑ | ↑ |
| **Larger Network size (covariant)** | ↓ | ↓ | ↓ |
| **Regularization (covariant)** | ↓ | ↓ | ↓ |

We present a unified framework for quantifying solution degeneracy in task-trained RNNs, identify the key factors that shape the solution landscape. In both machine learning and neuroscience, the optimal level of degeneracy may vary depending on the specific research questions being investigated.

This framework offers practical guidance for tailoring training to a given goal—whether encouraging consistency across models [59], or promoting diversity across learned solutions [60, 61, 62].

## 5 Acknowledgments

We acknowledge funding from NIH (RF1DA056403, U01NS136507), James S. McDonnell Foundation (220020466), Simons Foundation (Pilot Extension-00003332-02, McKnight Endowment Fund, CIFAR Azrieli Global Scholar Program, NSF (2046583), Harvard Medical School Neurobiology Lefler Small Grant Award, Harvard Medical School Dean's Innovation Award, Alice and Joseph Brooks Fund Postdoctoral Fellowship, and Kempner Graduate Fellowship. This work has been made possible in part by a gift from the Chan Zuckerberg Initiative Foundation to establish the Kempner Institute for the Study of Natural and Artificial Intelligence at Harvard University.

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

# Appendix

## A    Task descriptions

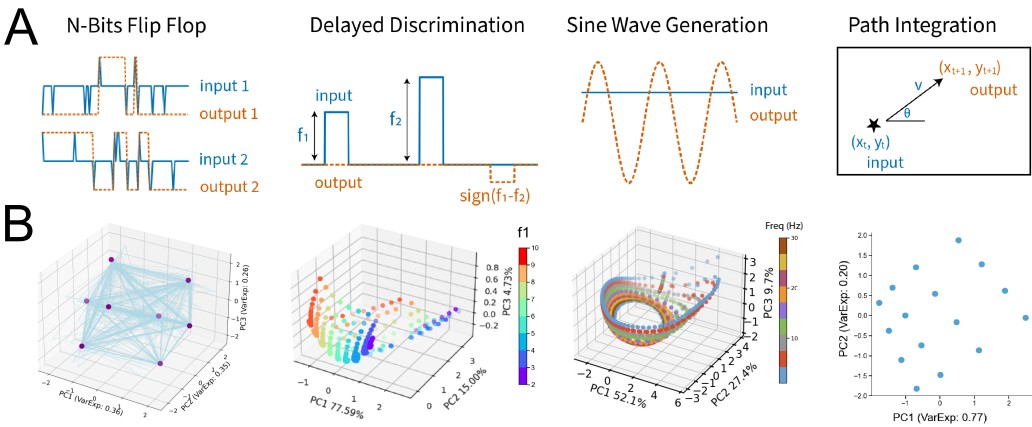

Figure 5: **Our task suite spans memory, integration, pattern generation, and decision-making.** Each task is designed to place distinct demands on the network's dynamics. **N-Bit Flip-Flop**: The network must remember the last nonzero input on each of $N$ independent channels. **Delayed Discrimination**: The network compares the magnitude of two pulses, separated by a variable delay, and outputs their sign difference. **Sine Wave Generation**: A static input specifies a target frequency, and the network generates the corresponding sine wave over time. **Path Integration**: The network integrates velocity inputs to track position in a bounded 2D or 3D arena (schematic shows 2D case).

**N-Bit Flip-Flop Task** Each RNN receives $N$ independent input channels taking values in $\{-1, 0, +1\}$, which switch with probability $p_{\text{switch}}$. The network has $N$ output channels that must retain the most recent nonzero input on their respective channels. The network dynamics form $2^N$ fixed points, corresponding to all binary combinations of $\{-1, +1\}^N$.

**Delayed Discrimination Task** The network receives two pulses of amplitudes $f_1, f_2 \in [2, 10]$, separated by a variable delay $t \in [5, 20]$ time steps, and must output $\text{sign}(f_2 - f_1)$. In the $N$-channel variant, comparisons are made independently across channels. The network forms task-relevant fixed points to retain the amplitude of $f_1$ during the delay period.

**Sine Wave Generation** The network receives a static input specifying a target frequency $f \in [1, 30]$ and must generate the corresponding sine wave $\sin(2\pi f t)$ over time. We define $N_{\text{freq}}$ target frequencies, evenly spaced within the range $[1, 30]$, and use them during training. In the $N$-channel variant, each input channel specifies a frequency, and the corresponding output channel generates a sine wave at that frequency. For each frequency, the network dynamics form and traverse a limit cycle that produces the corresponding sine wave.

**Path Integration Task** Starting from a random position in 2D, the network receives angular direction $\theta$ and speed $v$ at each time step and updates its position estimate. In the 3D variant, the network takes as input azimuth $\theta$, elevation $\phi$, and speed $v$, and outputs updated $(x, y, z)$ position. The network performs path integration by accumulating velocity vectors based on the input directions and speeds. After training, the network forms a Euclidean map of the environment in its internal state space.

## B    Additional axes of task complexity

In the main text, we controlled task complexity by varying the number of independent input–output channels, effectively duplicating the task across dimensions. Here, we explore two alternative approaches: increasing the task's memory demand and adding auxiliary objectives.

**Changing memory demand.** Of the four tasks, only Delayed Discrimination requires extended memory, as its performance depends on maintaining the first stimulus across a variable delay. See

Appendix H for a quantification of each task's memory demand. We increased the memory load in Delayed Discrimination by lengthening the delay period. This manipulation reduced degeneracy at the dynamical and behavioral levels but increased it at the weight level, mirroring the effect of increasing task dimensionality (Figure 6A).

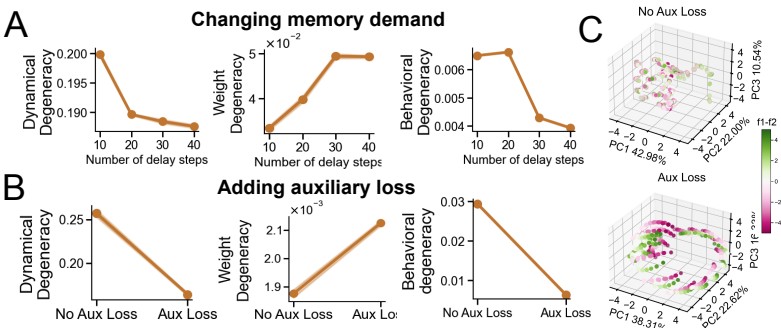

Figure 6: **Memory demand and auxiliary loss modulate degeneracy in distinct ways.** In the Delayed Discrimination task, both manipulations reduce dynamical and behavioral degeneracy while increasing weight degeneracy. The auxiliary loss also induces additional line attractors in the network's dynamics, as shown in (C).

**Adding auxiliary loss.** We next examined how adding an auxiliary loss affects solution degeneracy in the Delayed Discrimination task. Specifically, the network outputs both the sign and the magnitude of the difference between two stimulus values ($f_2 - f_1$), using separate output channels for each. This manipulation added a second output channel and increased memory demand by requiring the network to track the magnitude of the difference between incoming stimuli. Consistent with our hypothesis, this manipulation reduced dynamical and behavioral degeneracy while increasing weight degeneracy (Figure 6B). Crucially, the auxiliary loss induced additional line attractors in the network dynamics, further structuring internal trajectories and aligning neural responses across networks (Figure 6C). While the auxiliary loss increases both output dimensionality and temporal memory demand, we interpret its effect holistically as a structured increase in task complexity.

## C  Higher task complexity induces more feature learning

We hypothesize that the increased weight degeneracy observed in harder tasks reflects stronger feature learning. Specifically, harder tasks may force network weights to travel farther from their initialization. If more complex task variants, like those in Section 3.1, truly induce greater feature learning, then networks should traverse a greater distance in weight space, resulting in more dispersed final weights. To test this idea, we measured feature learning strength in networks trained on different task variants using two complementary metrics [39, 36]: **Weight-change norm:** $\|\mathbf{W}_T - \mathbf{W}_0\|_F$, where larger values indicate stronger feature learning. **Kernel alignment (KA):** measures the directional change of the neural tangent kernel (NTK) before and after training: $\mathrm{KA}\left(K^{(f)}, K^{(0)}\right) = \dfrac{\mathrm{Tr}\left(K^{(f)} K^{(0)}\right)}{\left\|K^{(f)}\right\|_F \left\|K^{(0)}\right\|_F}$, where $K = \nabla_W \hat{y}^\top \nabla_W \hat{y}$. Lower KA indicates greater NTK rotation and thus stronger feature learning.

More complex tasks consistently drive stronger feature learning and greater dispersion in weight space, as reflected by increasing weight-change norm and decreasing kernel alignment across all tasks (Figure 7).

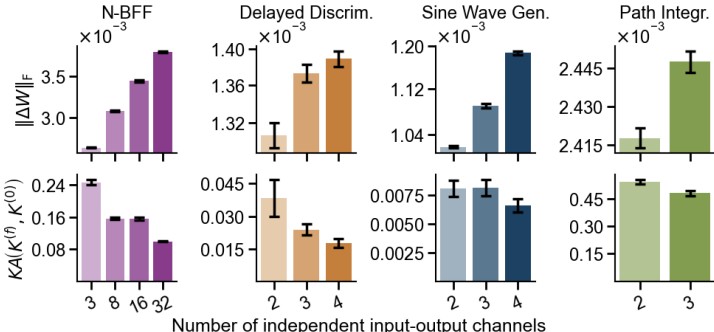

Figure 7: **More complex tasks drive stronger feature learning in RNNs.** Increased input–output dimensionality leads to higher weight-change norms and lower kernel alignment. Error bars indicate $\pm 1$ standard error.

## D   Feature learning effect for all tasks

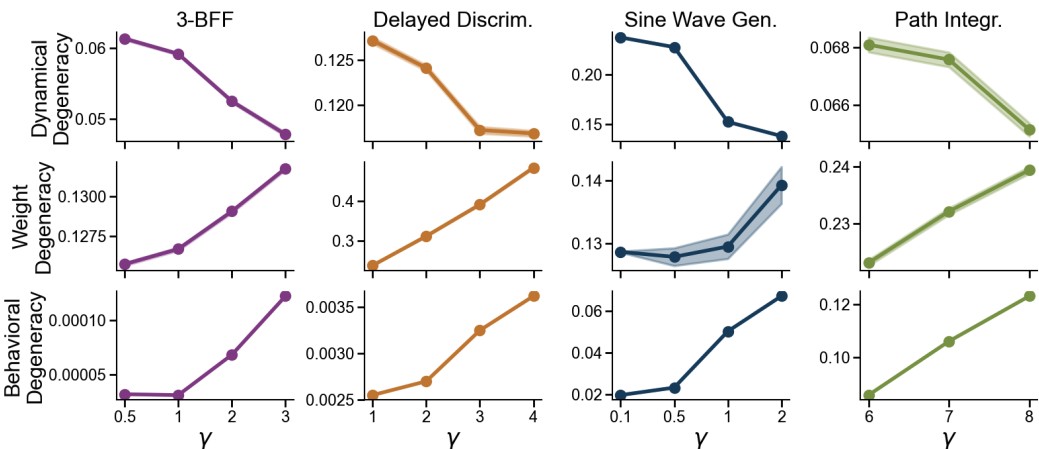

Figure 8: **Stronger feature learning reduces dynamical degeneracy but increases weight and behavioral degeneracy.** Panels show degeneracy at the dynamical, weight, and behavioral levels (top to bottom). Shaded area indicates $\pm 1$ standard error.

## E   Network size effect for all tasks

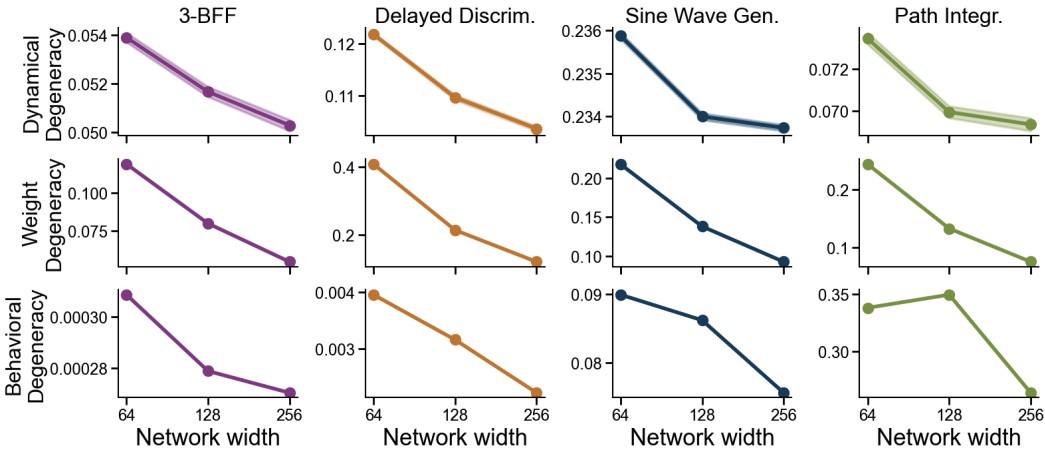

Figure 9: **Larger networks reduce degeneracy across weight, dynamics, and behavior.** Controlling for feature learning strength, wider RNNs yield more consistent solutions across all three levels of analysis. Panels show degeneracy at the dynamical, weight, and behavioral levels (top to bottom). Shaded area indicates $\pm 1$ standard error.

## F   Task details

### F.1   N-Bit Flip Flop

| Task Parameter | Value |
| --- | --- |
| Probability of flip | 0.3 |
| Number of time steps | 100 |

## F.2 Delayed Discrimination

| Task Parameter | Value |
| --- | --- |
| Number of time steps | 60 |
| Max delay | 20 |
| Lowest stimulus value | 2 |
| Highest stimulus value | 10 |

## F.3 Sine Wave Generation

| Task Parameter | Value |
| --- | --- |
| Number of time steps | 100 |
| Time step size | 0.01 |
| Lowest frequency | 1 |
| Highest frequency | 30 |
| Number of frequencies | 100 |

## F.4 Path Integration

| Task Parameter | Value |
| --- | --- |
| Number of time steps | 100 |
| Maximum speed ($v_{max}$) | 0.4 |
| Direction increment std ($\theta_{std}$ / $\phi_{std}$) | $\pi/10$ |
| Speed increment std | 0.1 |
| Noise std | 0.0001 |
| Mean stop duration | 30 |
| Mean go duration | 50 |
| Environment size (per side) | 10 |

# G Training details

## G.1 N-Bit Flip Flop

| Training Hyperparameter | Value |
| --- | --- |
| Optimizer | Adam |
| Learning rate | 0.001 |
| Learning rate scheduler | None |
| Max epochs | 300 |
| Steps per epoch | 128 |
| Batch size | 256 |
| Early stopping threshold | 0.001 |
| Patience | 3 |
| Time constant ($\mu P$) | 1 |

### G.2 Delayed Discrimination

| Training Hyperparameter | Value |
| --- | --- |
| Optimizer | Adam |
| Learning rate | 0.001 |
| Learning rate scheduler | CosineAnnealingWarmRestarts |
| Max epochs | 500 |
| Steps per epoch | 128 |
| Batch size | 256 |
| Early stopping threshold | 0.01 |
| Patience | 3 |
| Time constant ($\mu P$) | 0.1 |

### G.3 Sine Wave Generation

| Training Hyperparameter | Value |
| --- | --- |
| Optimizer | Adam |
| Learning rate | 0.0005 |
| Learning rate scheduler | None |
| Max epochs | 500 |
| Steps per epoch | 128 |
| Batch size | 32 |
| Early stopping threshold | 0.05 |
| Patience | 3 |
| Time constant ($\mu P$) | 1 |

### G.4 Path Integration

| Training Hyperparameter | Value |
| --- | --- |
| Optimizer | Adam |
| Learning rate | 0.001 |
| Learning rate scheduler | ReduceLROnPlateau |
| Learning rate decay factor | 0.5 |
| Learning rate decay patience | 40 |
| Max epochs | 1000 |
| Steps per epoch | 128 |
| Batch size | 64 |
| Early stopping threshold | 0.05 |
| Patience | 3 |
| Time constant ($\mu P$) | 0.1 |

## H  Memory demand of each task

In this section, we quantify each task's memory demand by measuring how far back in time its inputs influence the next output. Specifically, for each candidate history length $h$, we build feature vectors

$$\mathbf{s}_t^{(h)} = [\, x_{t-h+1}, \, \ldots, \, x_t; \, y_t \,] \; \in \; \mathbb{R}^{h\, d_{\text{in}} + d_{\text{out}}},$$

and **train a two-layer MLP to predict the subsequent target** $y_{t+1}$. We then evaluate the held-out mean-squared error $\text{MSE}(h)$, averaged over multiple random initializations. We identify the smallest history length $h^*$ at which the error curve plateaus or has a minimum, and take $h^*$ as the task's intrinsic memory demand.

From the results, we can see that the N-Bits Flip-Flop task requires only one time-step of memory—exactly what's needed to recall the most recent nonzero input in each channel. The Sine Wave Generation task demands two time-steps, reflecting the need to track both phase and direction of change. Path Integration likewise only needs one time-step, since the current position plus instantaneous velocity and heading suffice to predict the next position. Delayed Discrimination is the only memory-intensive task: our method estimates a memory demand of 25 time-steps, which happens to be the time interval between the offset of the first stimulus and the onset of the response period, during which the network needs to first keep track of the amplitude of the first stimulus and then its decision.

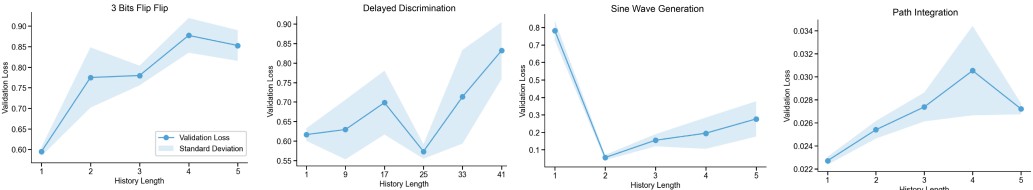

Figure 10: **Memory demand of each task**. The held-out mean-squared error $\text{MSE}(h)$ of a two-layer MLP predictor is plotted against history length $h$. The intrinsic memory demand $h^*$, defined by the plateau or minimum of each curve, is 1 for the N-Bits Flip-Flop and Path Integration tasks, 2 for Sine Wave Generation, and 25 for Delayed Discrimination—matching the inter-stimulus delay interval in that task.

# I  More details on the degeneracy metrics

## I.1  Dynamical Degeneracy

Briefly, DSA proceeds as follows: Given two RNNs with hidden states $\mathbf{h}_1(t) \in \mathbb{R}^n$ and $\mathbf{h}_2(t) \in \mathbb{R}^n$, we first generate a delay-embedded matrix, $\mathbf{H}_1$ and $\mathbf{H}_2$ of the hidden states in their original state space. Next, for each delay-embedded matrix, we use Dynamic Mode Decomposition (DMD) [63] to extract linear forward operators $\mathbf{A}_1$ and $\mathbf{A}_2$ of the two systems' dynamics. Finally, a Procrustes distance between the two matrices $\mathbf{A}_1$ and $\mathbf{A}_2$ is used to quantify the dissimilarity between the two dynamical systems and provide an overall DSA score, defined as:

$$d_{\text{Procrustes}}(\mathbf{A}_1, \mathbf{A}_2) = \min_{\mathbf{Q} \in O(n)} \|\mathbf{A}_1 - \mathbf{Q}\mathbf{A}_2\mathbf{Q}^{-1}\|_F$$

where $\mathbf{Q}$ is a rotation matrix from the orthogonal group $O(n)$ and $\|\cdot\|_F$ is the Frobenius norm. This metric quantifies how dissimilar the dynamics of the two RNNs are after accounting for orthogonal transformations. We quantify Dynamical Degeneracy across many RNNs as the average pairwise distance between pairs of RNN neural-dynamics (hidden-state trajectories).

After training, we extract each network's hidden-state activations for every trial in the training set, yielding a tensor of shape (trials × time steps × neurons). We collapse the first two dimensions and yield a matrix of size (trials × time steps) × neurons. We then apply PCA to retain the components that explain 99% of the variance to remove noisy and low-variance dimensions of the hidden state trajectories. Next, we perform a grid search over candidate delay lags, with a minimum lag of 1 and a maximum lag of 30, selecting the lag that minimizes the reconstruction error of DSA on the dimensionality reduced trajectories. Finally, we fit DSA with full rank and the optimal lag to these PCA-projected trajectories and compute the pairwise DSA distances between all networks.

## I.2  Weight degeneracy

We computed the pairwise distance between the recurrent matrices from different networks using Two-sided Permutation with One Transformation [64, 65] function from the Procrustes Python package [66].

# J   Representational degeneracy

We further quantified solution degeneracy at the representational level—that is, the variability in each network's internal feature space when presented with the same input dataset—using Singular Vector Canonical Correlation Analysis (SVCCA). SVCCA works by first applying singular value decomposition (SVD) to each network's activation matrix, isolating the principal components that capture most of its variance, and then performing canonical correlation analysis (CCA) to find the maximally correlated directions between the two reduced subspaces. The resulting canonical correlations therefore measure how similarly two networks represent the same inputs: high average correlations imply low representational degeneracy (i.e., shared feature subspaces), whereas lower correlations reveal greater divergence in what the models learn. We define the representational degeneracy (labeled as the SVCCA distance below) as

$$d_{\mathrm{repr}}(A_x, A_y) = 1 - \mathrm{SVCCA}(A_x, A_y).$$

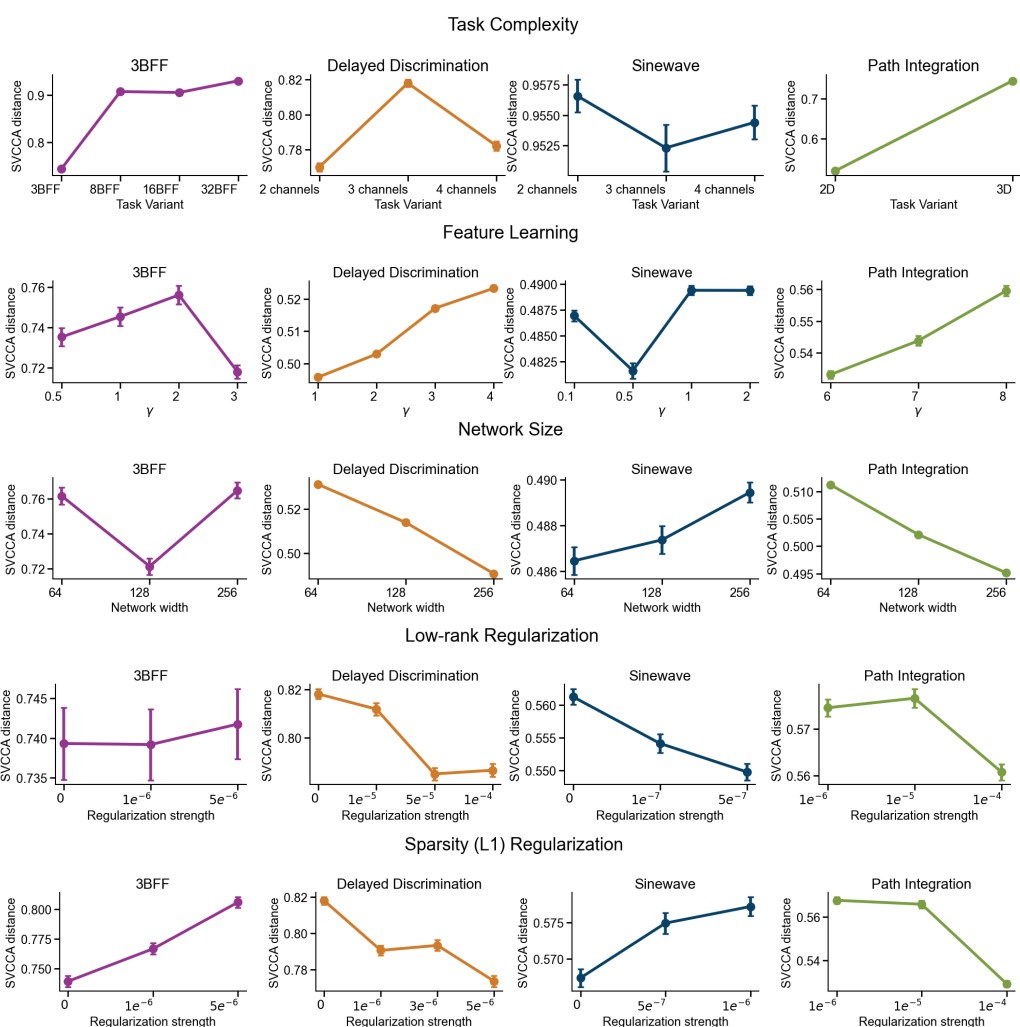

Figure 11: **Representational degeneracy, as measured by the average SVCCA distance between networks, does not necessarily change uniformly as we vary task complexity, feature learning strength, network size, and regularization strength.**

We found that as we vary the four factors that robustly control the dynamical degeneracy across task-trained RNNs, the representational-level degeneracy isn't necessarily constrained by those same factors in the same way. In RNNs, task-relevant computations are implemented at the level of

network's dynamics instead of static representations, and RNNs that implement similar temporal dynamics can have disparate representaional geometry. Therefore, it is expected that task complexity, learning regime, and network size change the task-relevant computations learned by the networks by affecting their neural dynamics instead of representations. DSA captures the dynamical aspect of the neural computation by fitting a forward operator matrix $A$ that maps the network's activity at one time step to the next, therefore directly capturing the temporal evolution of neural activities. By contrast, SVCCA aligns the principal subspaces of activation vectors at each time point but treats those vectors as independent samples—it never examines how one state evolves into the next. As a result, SVCCA measures only static representational similarity and cannot account for the temporal dependencies that underlie RNN computations. Nonetheless, we expect SVCCA might be more helpful in measuring the solution degeneracy in feedforward networks.

# K   Detailed characterization of OOD generalization performance

In addition to showing the behavioral degeneracy in the main text, here we provide a more detailed characterization of the OOD behavior of networks by showing the mean versus standard deviation, and the distribution of the OOD losses.

## K.1   Changing task complexity

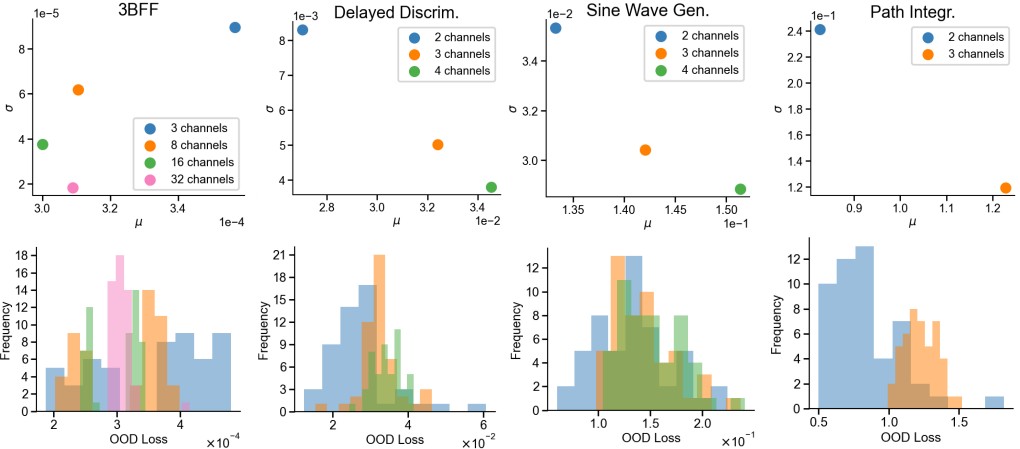

Figure 12: Detailed characterization of the OOD performance of networks while changing task complexity.

## K.2 Changing feature learning strength

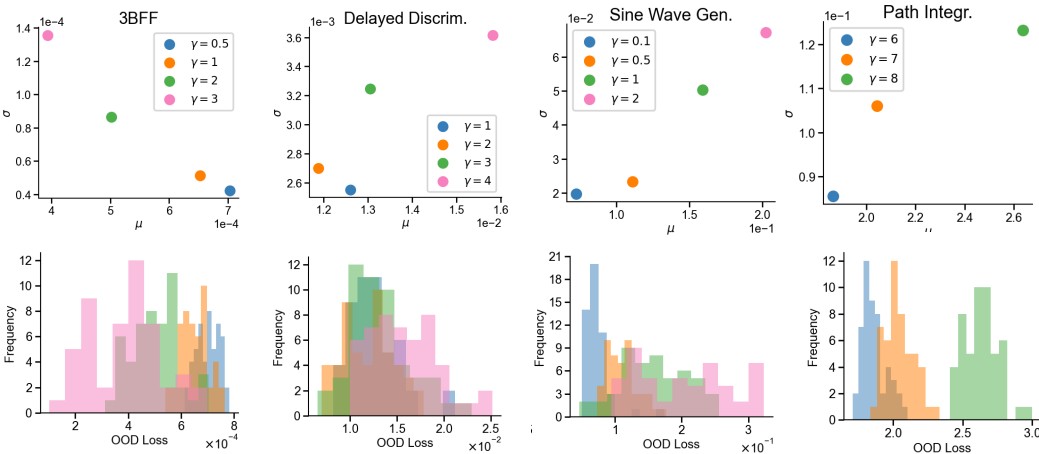

Figure 13: Detailed characterization of the OOD performance of networks while changing feature learning strength. Across Delayed Discrimination, Sine Wave Generation, and Path Integration tasks, networks trained with larger $\gamma$ – and thus undergoing stronger feature learning – exhibit higher mean OOD generalization loss together with higher variability, potentially reflecting overfitting to the training task.

## K.3 Changing network size

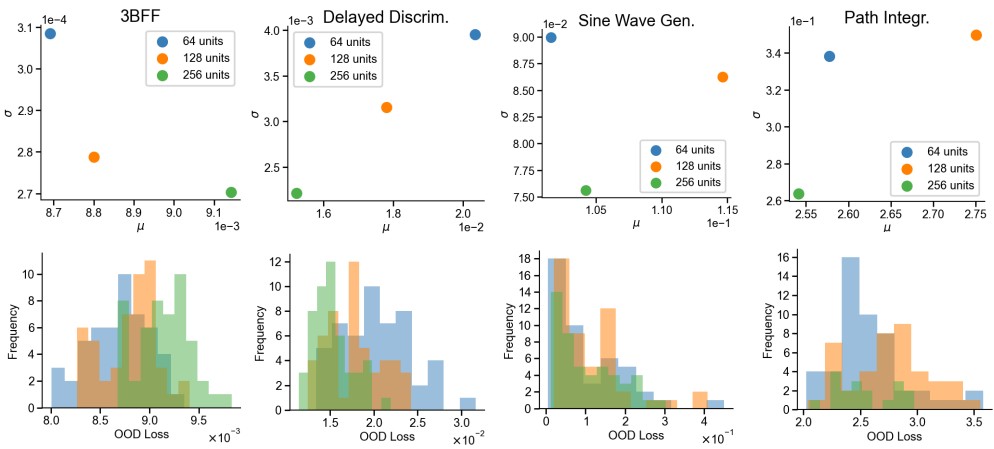

Figure 14: Detailed characterization of the OOD performance of networks while changing network size.

### K.4 Changing regularization strength

#### K.4.1 Low-rank regularization

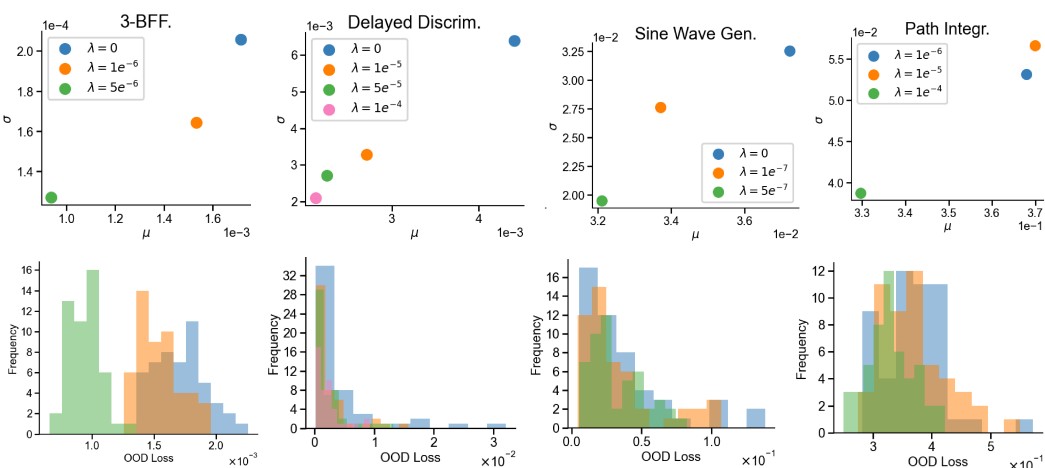

Figure 15: Detailed characterization of the OOD performance of networks while changing low-rank regularization strength.

#### K.4.2 Sparsity (L1) regularization

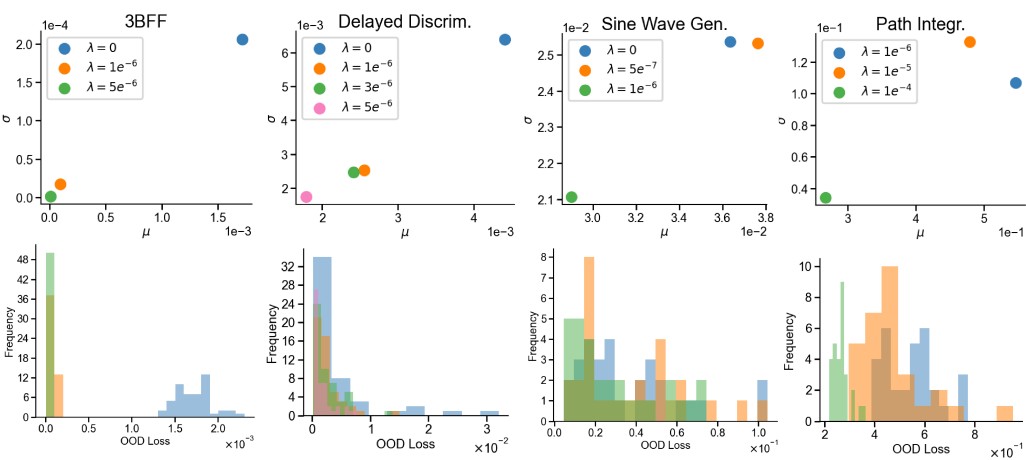

Figure 16: Detailed characterization of the OOD performance of networks while changing sparsity (L1) regularization strength.

## L   A short introduction to Maximal Update Parameterization ($\mu P$)

Under the NTK parametrization, as the network width goes to infinity, the network operates in the *lazy* regime, where its functional evolution is well-approximated by a first-order Taylor expansion around the initial parameters [49, 37, 32, 33]. In this limit feature learning is suppressed and training dynamics are governed by the fixed Neural Tangent Kernel (NTK).

To preserve non-trivial feature learning at large width, the *Maximal Update Parametrization* ($\mu$P) rescales both the weight initialisation and the learning rate. $\mu P$ keeps three quantities *width-invariant* at every layer—(i) the norm/variance of activations (ii) the norm/variance of the gradients, and (iii) the parameter updates applied by the optimizer [67, 68, 34, 35].

For recurrent neural networks, under Stochastic Gradient Descent (SGD), the network output, initialization, and learning rates are scaled as

$$f = \frac{1}{\gamma_0 N} \, \vec{w} \cdot \phi(h), \tag{1}$$

$$\partial_t h = -h + \frac{1}{\sqrt{N}} \, J \phi(h), \qquad J_{ij} \sim \mathcal{N}(0, 1), \tag{2}$$

$$\eta_{\text{SGD}} = \eta_0 \, \gamma_0^2 \, N. \tag{3}$$

Under Adam optimizer, the network output, initialization, and learning rates are scaled as

$$f = \frac{1}{\gamma_0 N} \, \vec{w} \cdot \phi(h), \tag{4}$$

$$\partial_t h = -h + \frac{1}{N} \, J \phi(h), \qquad J_{ij} \sim \mathcal{N}(0, N), \tag{5}$$

$$\eta_{\text{Adam}} = \eta_0 \, \gamma_0. \tag{6}$$

## M  Theoretical relationship between parameterizations

We compare two RNN formalisms used in different parts of the main manuscript: a standard discrete-time RNN trained with fixed learning rate and conventional initialization, and a $\mu$P-style RNN trained with leaky integrator dynamics and width-aware scaling.

In the standard discrete-time RNN, the hidden activations are updated as

$$h(t + 1) = \phi\big(W_h h(t) + W_x x(t)\big),$$

In $\mu P$ RNNs, the hidden activations are updated as

$$h(t + 1) - h(t) = \tau\big(-h(t) + \frac{1}{N} J \phi(h(t)) + U x(t)\big)$$

When $\tau = 1$,

$$h(t + 1) - h(t) = -h(t) + \frac{1}{N} J \phi(h(t)) + U x(t)$$

$$h(t + 1) = \frac{1}{N} J \phi(h(t)) + U x(t)$$

Aside from the overall scaling factor, the difference between the two parameterizations lies in the placement of the non-linearity:

- **Standard RNN:** $\phi$ is applied *post-activation*, i.e. after the recurrent and input terms are linearly combined,
- **$\mu$P RNN:** $\phi$ is applied *pre-activation*; i.e. before the recurrent weight matrix, so the hidden state is first non-linearized and then linearly combined

Miller and Fumarola [69] demonstrated that two classes of continuous-time firing-rate models which differ in their placement of the non-linearity are mathematically equivalent under a change of variables:

$$\text{v-model} \quad \tau \frac{dv}{dt} = -v + \tilde{I}(t) + W f(v)$$

$$\text{r-model:} \quad \tau \frac{dr}{dt} = -r + f(W r + I(t))$$

with equivalence holding under the transformation $v(t) = W r(t) + I(t)$ and $\tilde{I}(t) = I(t) + \tau \frac{dI}{dt}$, assuming matched initial conditions.

Briefly, they show that $W r + I$ evolves according to the $v$-equation as follows:

$$v(t) = Wr(t) + I(t)$$

$$\frac{dv}{dt} = \frac{d}{dt}\big(Wr(t) + I(t)\big)$$

$$= W\frac{dr}{dt} + \frac{dI}{dt}$$

$$= W\left(\frac{1}{\tau}\left(-r + f(Wr + I)\right)\right) + \frac{dI}{dt}$$

$$\tau\frac{dv}{dt} = -Wr + Wf(Wr + I) + \tau\frac{dI}{dt}$$

$$= -(v - I) + Wf(v) + \tau\frac{dI}{dt}$$

$$= -v + I + \tau\frac{dI}{dt} + Wf(v)$$

$$\tau\frac{dv}{dt} = -v + \tilde{I}(t) + Wf(v)$$

This mapping applies directly to RNNs viewed as continuous-time dynamical systems and helps relate v-type $\mu$P-style RNNs to standard discrete-time RNNs. It suggests that the $\mu$P RNN (in v-type form) and the standard RNN (in r-type form) can be treated as different parameterizations of the same underlying dynamical system when:

- Initialization scales are matched
- The learning rate is scaled appropriately with $\gamma$
- Output weight norms are adjusted according to width

In summary, while a theoretical equivalence exists, it is contingent on consistent scaling across all components of the model. In this manuscript, we use the standard discrete-time RNNs due to its practical relevance for task-driven modeling community, while switching to $\mu$P to isolate the effect of feature learning and network size. Additionally, we confirm that the feature learning and network size effects on degeneracy hold qualitatively the same in standard discrete-time RNNs, unless where altering network width induces unstable and lazier learning in larger networks (Figure Q and R).

# N Verifying larger $\gamma$ reliably induces stronger feature learning in $\mu P$

In $\mu P$ parameterization, the parameter $\gamma$ interpolates between lazy training and rich, feature-learning dynamics, without itself being the absolute magnitude of feature learning. Here, we assess feature-learning strength in RNNs under varying $\gamma$ using two complementary metrics:

**Weight-change norm** which measures the magnitude of weight change throughout training. A larger weight change norm indicates that the network undergoes richer learning or more feature learning.

$$\frac{\|\mathbf{W}_T - \mathbf{W}_0\|_F}{N},$$

where N is the number of parameters in the weight matrices being compared.

**Kernel alignment (KA)**, which measures the directional change of the neural tangent kernel (NTK) before and after training. A lower KA score corresponds to a larger NTK rotation and thus stronger feature learning.

$$\text{KA}\big(K^{(f)}, K^{(0)}\big) \; = \; \frac{\text{Tr}\big(K^{(f)} K^{(0)}\big)}{\big\|K^{(f)}\big\|_F \big\|K^{(0)}\big\|_F}, \qquad K \; = \; \nabla_W \hat{y}^\top \nabla_W \hat{y}.$$

We demonstrate that higher $\gamma$ indeed amplifies feature learning inside the network.

## N.1 N-BFF

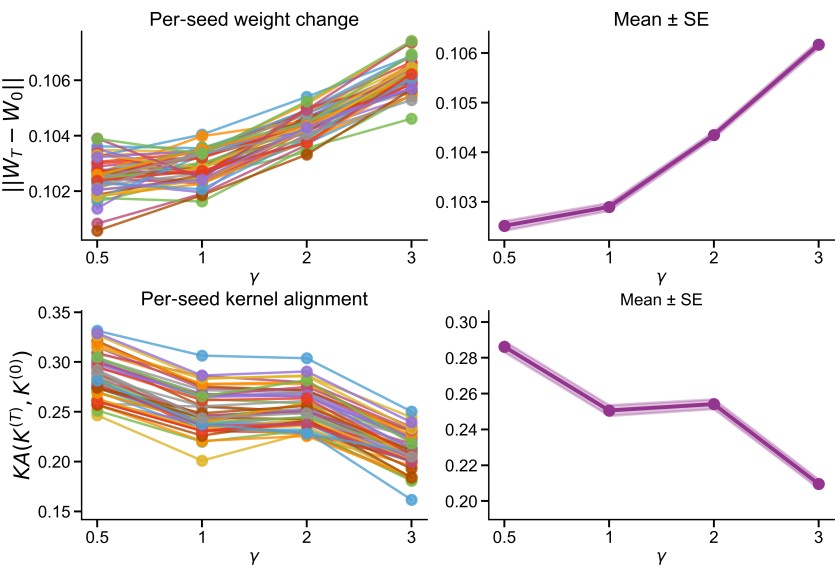

Figure 17: Weight change norm and kernel alignment for networks trained on the 3-Bits Flip Flop task as we vary $\gamma$. On the left panels, we show the per-seed metrics where connected dots of the same color are networks of identical initialization trained with different $\gamma$. On the right panels, we show the mean and standard error of the metrics across 50 networks. For larger $\gamma$, the weights move further from their initializations as shown by the larger weight change norm, and their NTK evolves more distinct from the network's NTK at initialization as shown by the reduced KA. Both indicate stronger feature learning for networks trained under larger $\gamma$.

## N.2 Delayed Discrimination

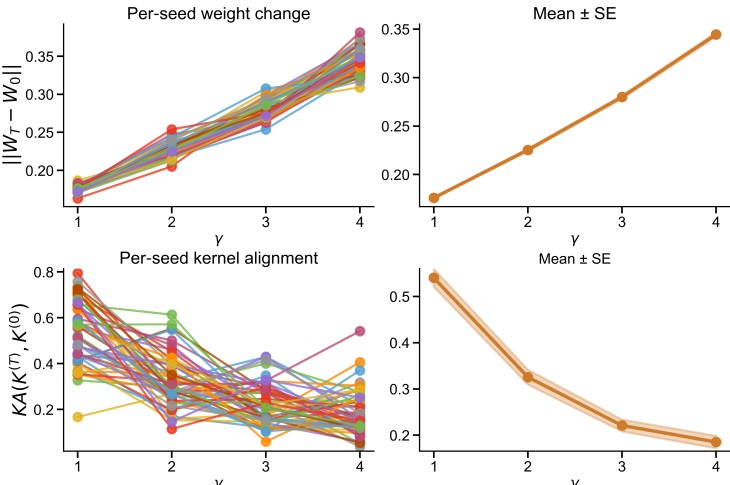

Figure 18: Stronger feature learning for networks trained under larger $\gamma$ on the Delayed Discrimination task.

## N.3 Sine Wave Generation

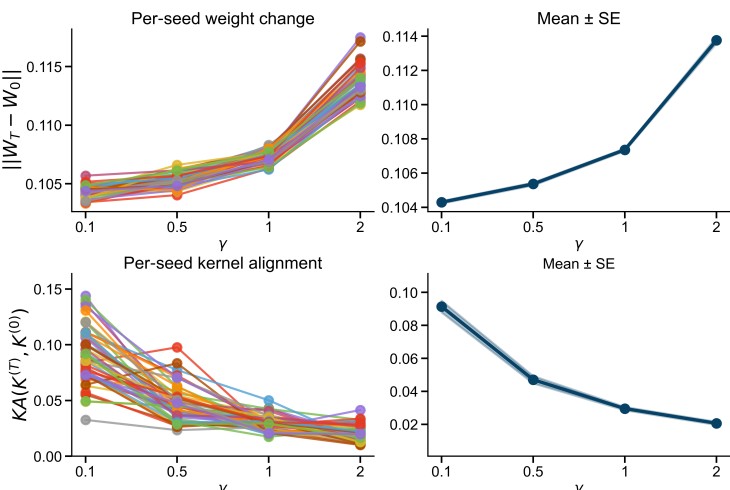

Figure 19: Stronger feature learning for networks trained under larger $\gamma$ on the Sine Wave Generation task.

## N.4 Path Integration

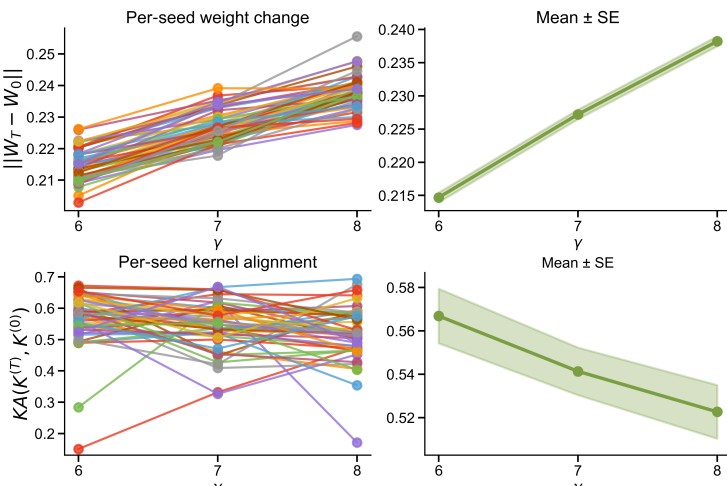

Figure 20: Stronger feature learning for networks trained under larger $\gamma$ on the Path Integration task.

## O Verifying $\mu P$ reliably controls for feature learning across network width

Here, we only use Kernel Alignment to assess the feature learning strength in the networks since the unnormalized weight-change norm $\|\mathbf{W}_T - \mathbf{W}_0\|_F$ scales directly with matrix size (therefore network size) and there exists no obvious way to normalize across different dimensions. In our earlier analysis where we compared weight-change norms at varying $\gamma$, network size remained fixed, so those Frobenius-norm measures were directly comparable. We found that, for all tasks except Delayed Discrimination, the change in mean KA across different network sizes remains extremely small (less than 0.1), which demonstrates that $\mu P$ parameterization with the same $\gamma$ has effectively controlled for feature learning strength across network sizes. On Delayed Discrimination, the networks undergo slightly lazier learning for larger network sizes. Nevertheless, we still include Delayed Discrimination in our analyses of solution degeneracy to ensure *our conclusions remain robust even when $\mu P$ can't perfectly equalize feature-learning strength across widths.* As shown in the main paper, lazier learning regime generally increases dynamical degeneracy; yet, larger networks which exhibit lazier learning in the N-BFF task actually display lower dynamical degeneracy. This reversed trend confirms that the changes in solution degeneracy arise from network size itself, not from residual variation in feature learning strength.

### O.1 N-BFF

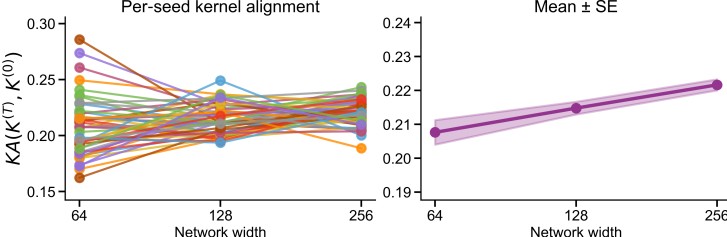

Figure 21: Kernel alignment (KA) for different network width on the 3 Bits Flip-Flop task. (Lower KA implies more feature learning.)

## O.2 Delayed Discrimination

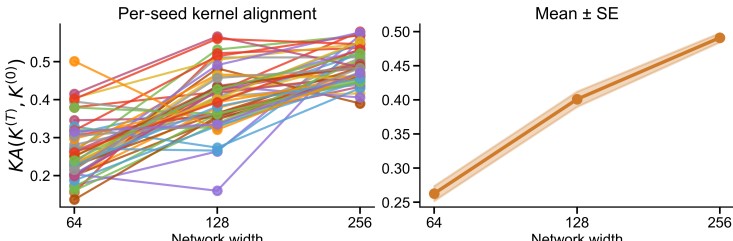

Figure 22: Kernel alignment for different network width on the Delayed Discrimination task.

## O.3 Sine Wave Generation

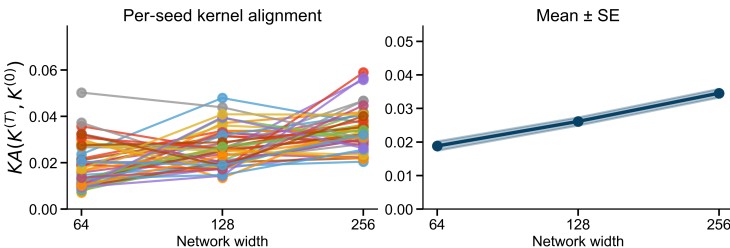

Figure 23: Kernel alignment for different network width on the Sine Wave Generation task.

## O.4 Path Integration

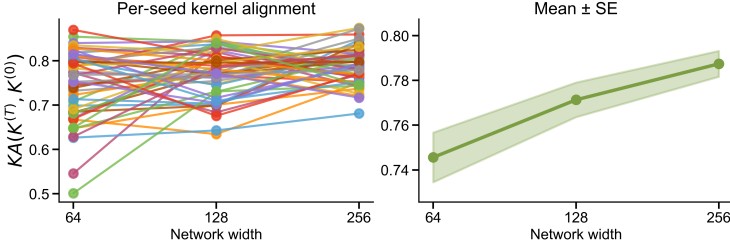

Figure 24: Kernel alignment for different network width on the Path Integration task.

# P Regularization's effect on degeneracy for all tasks

In addition to showing regularization's effect on degeneracy in Delayed Discrimination task in the main paper, here we show that heavier low-rank regularization and sparsity regularization also reliably reduce solution degeneracy across neural dynamics, weights, and OOD behavior in the other three tasks.

## P.1 Low-rank regularization

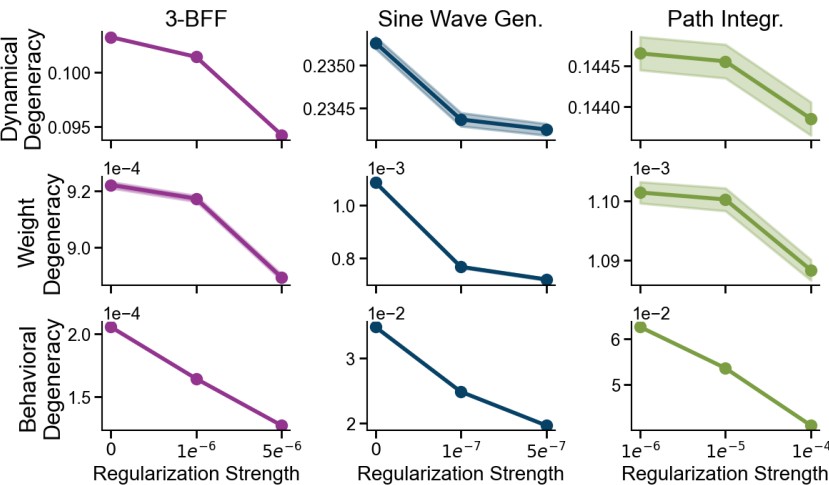

Figure 25: Low-rank regularization reduces degeneracy across neural dynamics, weight, and OOD behavior on the N-BFF, Sinewave Generation, and Path Integration task.

## P.2 Sparsity regularization

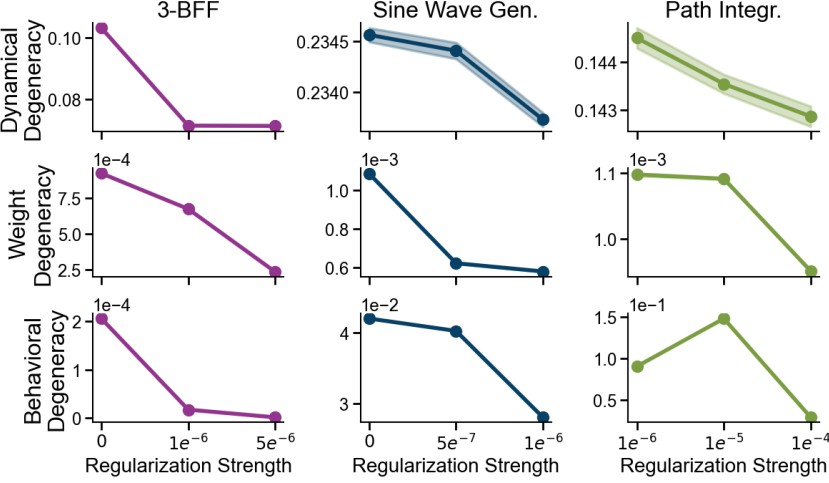

Figure 26: Sparsity regularization reduces degeneracy across neural dynamics, weight, and OOD behavior on the N-BFF, Sinewave Generation, and Path Integration task.

## Q Test feature learning effect on degeneracy in standard parameterization

While $\mu P$ lets us systematically vary feature-learning strength to study its impact on solution degeneracy, we confirm that the same qualitative pattern appears in *standard* discrete-time RNNs: stronger feature learning **lowers dynamical degeneracy** and **raises weight degeneracy** (Figure 27).

To manipulate feature-learning strength in these ordinary RNNs we applied the $\gamma$-**trick**—scaling the network's outputs by $\gamma$—and multiplied the learning rate by the same factor. With width fixed, these two operations replicate the effective changes induced by $\mu P$. Figure 28 shows that this combination reliably tunes feature-learning strength. Besides weight-change norm and kernel alignment, we also report **representation alignment (RA)**, giving a more fine-grained view of how much the learned features deviate from their initialization [39]. Representation alignment is the directional change of the network's represenational dissimilarity matrix before and after training, and is defined by

$$\mathrm{RA}\big(R^{(T)}, R^{(0)}\big) := \frac{\mathrm{Tr}\big(R^{(T)} R^{(0)}\big)}{\|R^{(T)}\| \, \|R^{(0)}\|}, \qquad R := H^{\top} H,$$

A lower RA means more change in the network's representation of inputs before and after training, and indicates stronger feature learning.

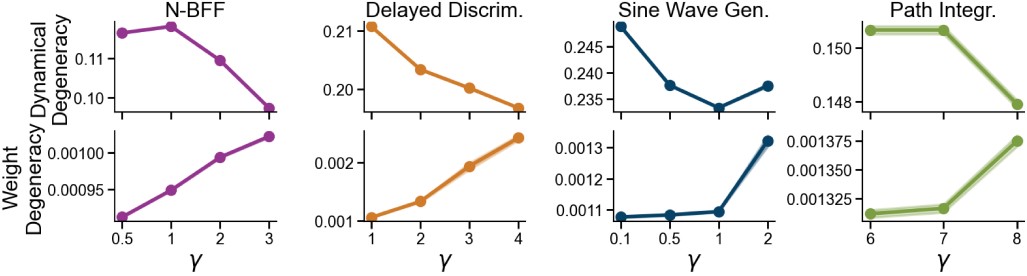

Figure 27: Stronger feature learning reliably decreases dynamical degeneracy while increasing weight degeneracy in standard discrete-time RNNs.

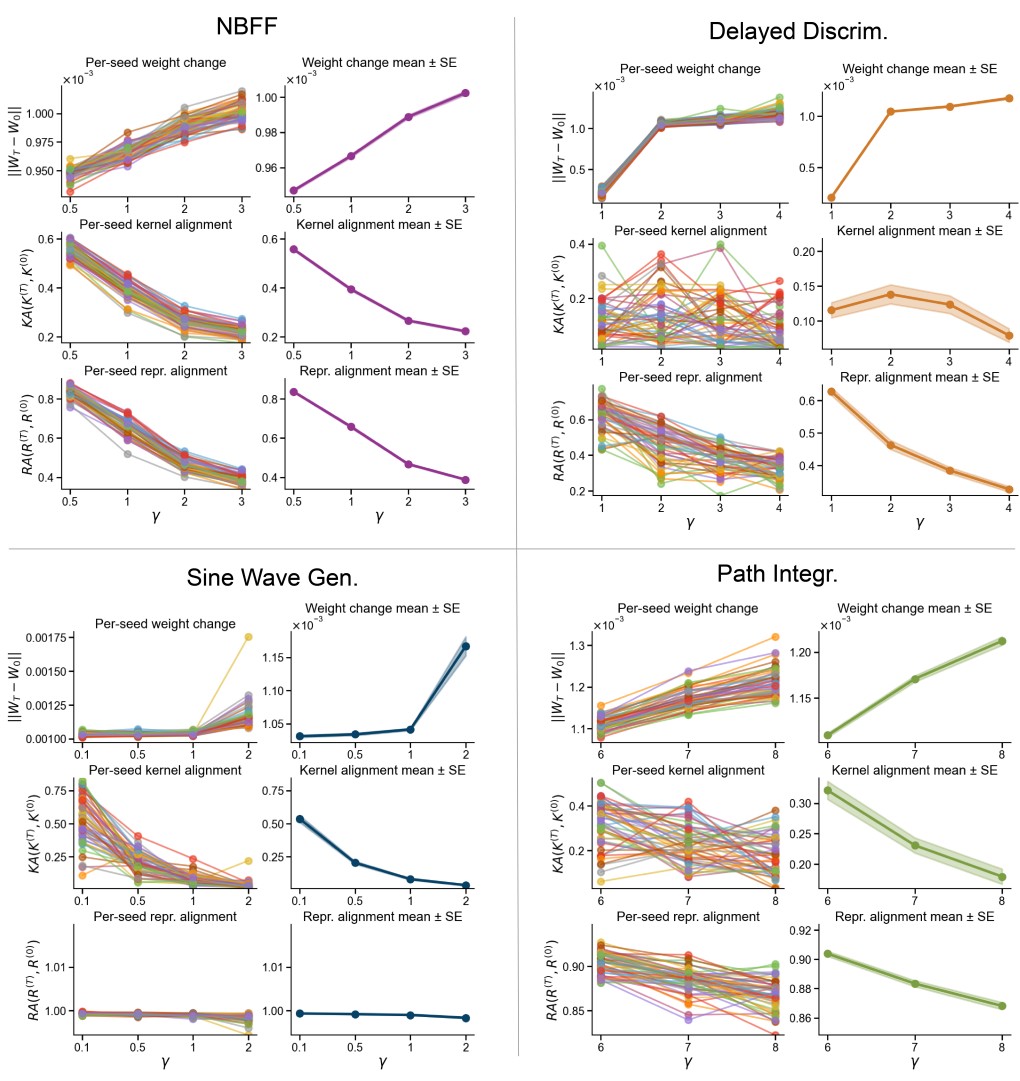

Figure 28: Larger $\gamma$ reliably induces stronger feature learning in standard discrete-time RNNs.

# R    Test network size effect on degeneracy in standard parameterization

When we vary network width, both the standard parameterization and $\mu P$ parameterization display the same overall pattern: **larger networks exhibit lower dynamical and weight degeneracy**. An exception arises in the 3BFF task, where feature learning becomes unstable and collapses in the wider models. In that setting we instead see *higher* dynamical degeneracy, which we suspect because the feature learning effect (lazier learning leads to higher dynamical degeneracy) dominates the network size effect.

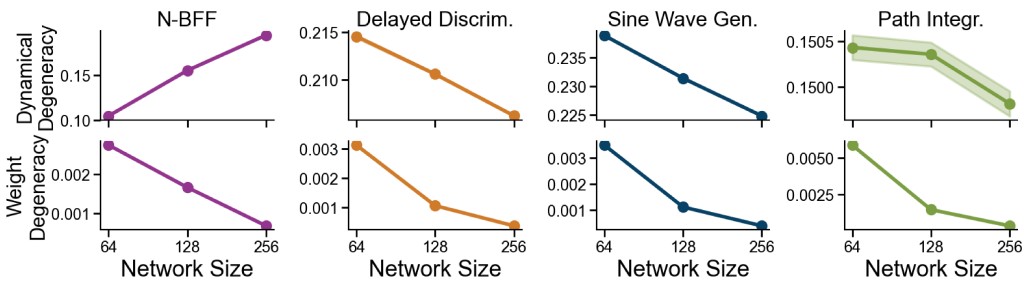

Figure 29: Larger network sizes lead to lower dynamical and weight degeneracy, except in the case where feature learning is unstable across width (in N-BFF).

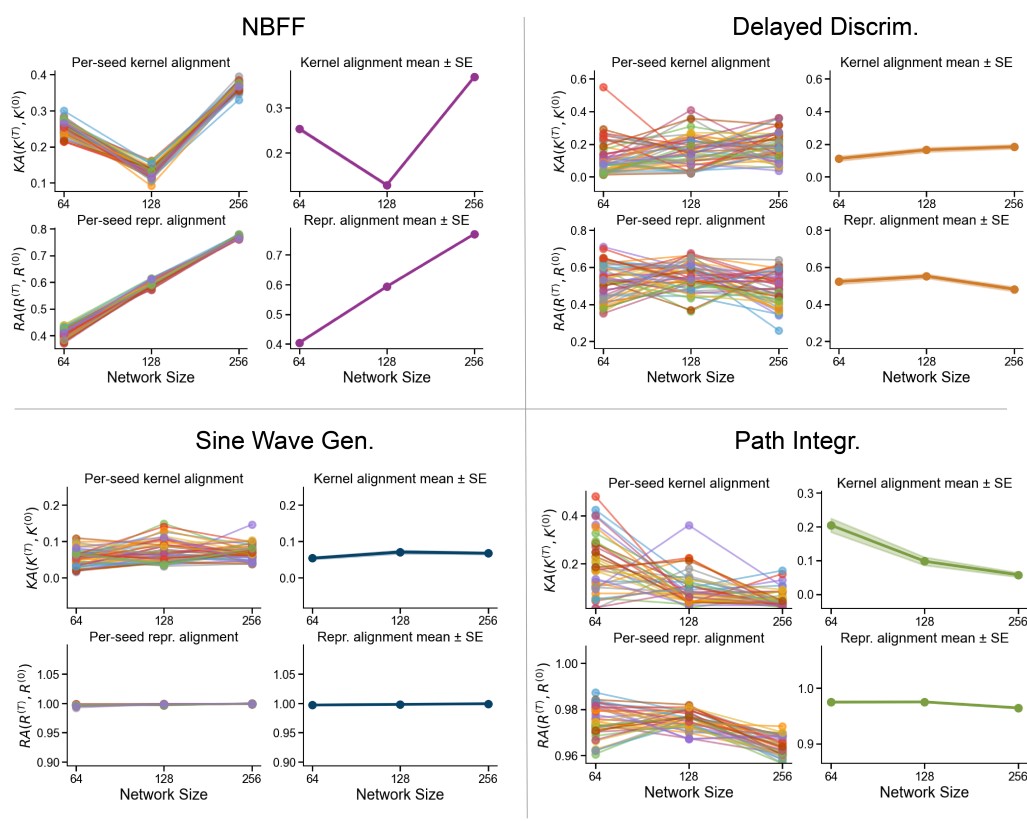

Figure 30: When changing network width in standard discrete-time RNNs, feature learning strength remains stable across width except in N-BFF, where notably lazier learning happens in the widest network.

# S Disclosure of compute resources

In this study, we conducted 50 independent training runs on each of four tasks, systematically sweeping four factors that modulate solution degeneracy—task complexity (15 experiments), learning regime (15 experiments), network size (12 experiments), and regularization strength (26 experiments), resulting in a total of 3400 networks. Each experiment was allocated 5 NVIDIA V100/A100 GPUs, 32 CPU cores, 256 GB of RAM, and a 4-hour wall-clock limit, for a total compute cost of approximately 68 000 GPU-hours.

