# OpenReview forum: "Measuring and Controlling Solution Degeneracy across Task-Trained Recurrent Neural Networks"
_NeurIPS.cc/2025/Workshop/UniReps — UniReps2025_

### Official Review · Reviewer_Sd3A · 2025-09-10
**Novel framework with solid methodology, but unclear rationale for some choices, and limited evidence and scope for some claims**

**Confidence:** 3

**Review:**

Summary:

The work introduces a framework of factors that affect different types of solution degeneracy in Recurrent Neural Networks (RNNs). The authors benchmark it using four tasks that represent various cognitive functions in neuroscience. Through systematic test cases that modify network hyperparameters, size, regularization, and task complexity, as measured by out-of-distribution (OOD) error, Dynamical Similarity Analysis (DSA), and permutation-invariant Frobenius (PIF) distance between recurrent weights, the authors determine which factors are covariant and contravariant to test the recent Contravariance Principle hypothesis. They also explore the relationships between task complexity and network width with feature learning and study representational degeneracy via Singular Vector Canonical Correlation Analysis (SVCCA).

Strengths:

Despite the limited scope inherent to an extended abstract, the work offers strong theoretical insights into solution degeneracy in task learning.

The study brings together different modern metrics (like DSA) and model modifications (like Maximal Update Parametrization) in a single framework. It is careful to separate correlated factors, such as network width and the strength of feature learning.

Detailed information is provided regarding hyperparameters and training regimes for replication, and repetitions were performed to test for variability.

The mathematics of the methods involved is discussed in detail, and sufficient references were provided to reconstruct the line of reasoning.

Weaknesses:

The rationale behind the choice of task types and examples is unclear. Were they selected based on benchmarks such as NeuroGym, or derived from other criteria?

https://pure.eur.nl/ws/portalfiles/portal/59119331/MolanoMazon_Neurogym.pdf

There may be a missed opportunity to utilize the recent discovery that the alignment of model weights may be measured using covariances:

https://arxiv.org/abs/2409.19460.

The claim that task difficulty depends on input/output size or memory demand seems reductive. For example, Sudoku puzzles with more given values can be more difficult than those with fewer given values. Similarly, memorization tasks require a lot of memory but are conceptually simple. They are more challenging in the context of RNNs, which simulate the retention of memory over time. However, the paper introducing the contravariance principle envisions it more broadly.

https://www.researchgate.net/publication/220804141_Construction_of_Heuristics_for_a_Search-Based_Approach_to_Solving_Sudoku

The study also does not test on more modern recurrence models, such as LSTMs, GRUs, and Mamba.

Comments:

Since this is an extended abstract and therefore a work in progress, I have rated it a 4. However, had it been a proceedings paper, which it could have been rewritten as, given its 31-page length, including the appendix, I think reviewers would have rated it an average of 3. I hope my questions and suggestions provided in the weaknesses section will help polish the paper.

**Score:**

4

**Topic Fit:**

3

---

### Official Review · Reviewer_cmAs · 2025-09-12
**A unified framework for quantifying and controlling solution degeneracy in task-trained RNNs**

**Confidence:** 5

**Review:**

## Summary
The paper introduces a framework to measure and manipulate solution degeneracy in RNNs trained on neuroscience-relevant tasks, examining three levels of behavior, neural dynamics, and weight levels. The framework performance is illustrated on 3,400 RNNs across four tasks to investigate how factors like task complexity, feature learning strength, network size, and structural regularization influence degeneracy, validating the contravariance principle and providing guidance for interpretable modeling.

## Main contribution of the paper:
- A unified framework for quantifying degeneracy at three levels
- The contravariance principle was validated by showing that higher task complexity reduces dynamical and behavioral degeneracy but increases weight degeneracy.
- They demonstrated that stronger feature learning decreases dynamical degeneracy, but on the other hand increases weight and behavioral degeneracy due to over fitting.
- Illustrated that large network widths given that feature learning is controlled and low-rank structural regularization reduce degeneracy across all levels.

## Strengths:
- The provided framework provides a multi-level quantification of degeneracy which enables analysis and testing of a theoretical principle like contravariance.

## Weaknesses:
- Limitation to vanilla RNNs only.

Overall, this contribution provides initial promising results, despite the limitations to only one type of RNNs.

**Score:**

2

**Topic Fit:**

3

---

### Official Review · Reviewer_hEMR · 2025-09-16
**Empirical analysis of degeneracy in RNNs**

**Confidence:** 4

**Review:**

Summary:

The paper studies empirical dynamics of RNNs, focusing on solution degeneracy. The authors test the contravariance principle, which suggests that as tasks become more complex, networks must reduce degeneracy and learn more structured internal representations. Through many experiments, they examine key factors that shape different degeneracy types.

Strengths:

- Very careful and systematic experimental design.
- Results are clearly presented and broadly consistent with theoretical expectations.

Weaknesses (minor):

- Some findings are expected (e.g., task complexity narrowing the solution space in fixed-capacity networks, or certain reparameterizations driving stronger feature learning thus less degeneracy), but the authors add value by systematically quantifying these effects.

**Score:**

4

**Topic Fit:**

3

---

### Official Review · Reviewer_rWRd · 2025-09-16
**Solution degeneracy in task-trained RNNs**

**Confidence:** 4

**Review:**

## Summary

The paper studies the phenomenon of solution degeneracy in task-trained RNNs, i.e. the fact that RNNs trained to solve the same task may exhibit widely different solutions. The authors study the degeneracy of solutions across three different levels:  "behavior", learned dynamics, and weight configurations.

The authors systematically characterize degeneracy of solutions in RNNs trained in four neuroscience tasks: pattern recognition
(N-Bit Flip-Flop), delayed decision-making (Delayed Discrimination), pattern generation (Sine Wave
Generation), and evidence accumulation (Path Integration) by varying task complexity, learning
regime (through weight matrix initialization), and network width. They further study the impact of structural regularization on the resulting solution degeneracy (imposing sparseness or low-rankness).


The authors report that increasing task complexity and using smaller initial weight magnitudes, which encourage stronger feature learning, lead to more consistent solutions at the level of dynamics, but less consistency in terms of weight configurations, while the effects at the behavioral level are mixed. The finding of greater variability in weight configurations was not entirely straightforward from my perspective, but the authors provide a convincing explanation: more complex tasks correspond to a more dispersed set of local minima in the weight space, each representing an equally good solution.

Overall the paper is an interesting contribution for the workshop and is clearly related to the workshop's themes. The critiques below aim  to improve the work and its contributions.

- I am a bit skeptical about the chosen way to quantify degeneracy at the behavioral level. The authors quantify behavioural degeneracy" by computing the variability in responses across networks when tested on out-of-distribution inputs (OOD). OOD inputs are defined by doubling the delay for the Delayed Discrimination task
and doubling the trial length for other tasks.
From my perspective, the “behavior” of an RNN is best captured by the strategy it employs to solve the task, which is inherently tied to its dynamical landscape. Thus,  I would quantify behavioral variability by testing multiple RNNs on the exact same sequence of trials and then measuring correlations of their outputs, for example, whether all models fail on the same trials, or whether different models fail on distinct types of trials due to different underlying strategies. This approach would be particularly informative if networks are not perfectly trained or tested during training (see also Liebana et al., 2025). For perfectly trained networks, however, comparing their dynamical landscapes might provide a more direct characterization of behavioral strategy similarity.

- For weight degeneracy I would think that would be also informative to compare the spectral properties of the weight matrices, e.g., eigenvalue distributions/ outliers, effective rank, and alignment of leading eigenvectors. I think that solutions with qualitatively different spectra would provide the first stage of weight degeneracy.

- I find the concept of task complexity somewhat overloaded and not clearly defined, though it remains an interesting and important open question. The authors do a commendable job of approaching it from multiple angles by increasing the number of input/output associations, or the memory demands of the task, and by introducing additional output requirements in the Delayed Discrimination task (where the network must report both the sign and magnitude of the two inputs). However, when reporting results, I would suggest being more explicit: instead of framing findings as “changing task complexity led to…,” it would be more prudent to state which specific factor of complexity was manipulated and describe the observed effect in those terms.

- The paper presents a systematic quantification of factors influencing variability of solutions across RNN training runs. Still, it would be important to explicitly perform experiments to control for sources of randomness such as initial conditions conditions of the state variables, and trial shuffling to separate their contributions from contributions of the variables the authors manipulate.


## Minor

- I find it a bit strange that the authors chose to call their work a “framework”. From my perspective the paper is a very interesting exploratory analysis on solution degeneracy of RNNs, but I wouldn't call it a framework.

- In line 39-40: I think it would read better if you mention the key factors, i.e. task complexity, learning
regime, network width, and structural regularization, in line 39, and drop the sentence in line 40. Then on 41: “By systematically varying these factors,…”

**Score:**

4

**Topic Fit:**

2